# Maternal Sleep Problems in the Periconceptional Period and the Impact on Health of Mother and Offspring: A Systematic Review

**Babette Bais** [1], **Milan G. Zarchev** [2], **Annemarie I. Luik** [3], **Lenie van Rossem** [1] **and Régine P. M. Steegers-Theunissen** [1,*]

1    Department of Obstetrics and Gynaecology, Erasmus University Medical Center,
     3015 GD Rotterdam, The Netherlands
2    Department of Psychiatry, Erasmus University Medical Center, 3015 GD Rotterdam, The Netherlands
3    Department of Epidemiology, Erasmus University Medical Center, 3015 GD Rotterdam, The Netherlands
*    Correspondence: r.steegers@erasmusmc.nl; Tel.: +31-612472643

**Abstract:** Knowledge of the impact of sleep problems in the periconceptional period is scarce. Since this period is the most sensitive time window for embryonic and placental development, we aim to study the impact of maternal sleep problems in the periconceptional period on both mother and offspring. We systematically searched various databases up until September 2021 for studies reporting on maternal sleep in the periconceptional period and any outcome in mother and offspring. We included observational studies describing maternal sleep problems in the periconceptional period and associations with either maternal and/or offspring outcomes. The search produced 8596 articles, of which we selected 27 studies. Some associations were found between sleep problems and lower fertility, more hypertensive disorders, more mood disorders in mothers, higher risk of preterm birth and low birth weight, and more sleep and behavior problems in offspring, with associations with maternal mood disorders being most consistent. This systematic review shows that maternal sleep problems in the periconceptional period are associated with a higher risk of various adverse outcomes in both mother and offspring, although not consistently. It shows that good sleep during pregnancy is crucial, starting as early as before conception, especially for maternal mood. Therefore, it is important for clinicians to pay attention to sleep problems in the periconceptional period and provide adequate treatment for potential sleep problems, even before pregnancy.

**Keywords:** sleep; insomnia; pregnancy; preconception; first trimester; periconception

## 1. Introduction

In approximately half of all pregnancies, women experience significant sleep problems, especially in the third trimester [1]. These problems may not only affect the health of the mother and offspring during pregnancy but also across their life course. Maternal sleep problems are associated with various maternal adverse outcomes, such as a higher risk of excessive weight gain (odds ratio (OR) 3.47, 95% confidence interval (CI) 1.25–9.62) [2] and gestational diabetes (OR 1.77, 95% CI 1.20–2.61) [3]. Poor maternal sleep during pregnancy is also associated with a higher risk of adverse outcomes at birth and in the offspring, such as preterm birth (relative risk (RR) 1.54, 95% CI 1.18–2.01) [4], being small for gestational age (OR 1.4, 95% CI 1.1–1.9) [5], and higher blood pressure in the child (β 1.6 mmHg, 95% CI 0.5–2.7) [6].

The impact of poor sleep is often studied during late pregnancy, but to a lesser extent in the periconceptional period, defined as 14 weeks before up until 10 weeks after conception [7]. This period is the most sensitive time window in the developmental origin of adverse reproductive outcomes, e.g., subfertility, miscarriage, congenital anomalies, low birth weight, but also of diseases later in life, such as hypertensive disorders and diabetes [7].

Although this period has largely been neglected in patient care and research, there is evidence that periconceptional sleep problems are also associated with various adverse outcomes, such as subfertility (hazard ratio (HR) 3.72, 95% CI 2.16–6.41) [8], an increased risk of spina bifida (OR 4.1, 95% CI 1.9–8.8) [9], and of simple and severe congenital heart disease (OR 2.5, 95% CI 1.6–3.8, and OR 2.0, 95% CI 1.3–3.0, respectively) [10]. However, an overview of the impact of periconceptional sleep problems is lacking.

In this systematic review, we aim to present an extensive overview of all studies that have been conducted on maternal sleep problems in the periconceptional period in association with various maternal and offspring outcomes.

## 2. Results

### 2.1. Study Selection

The literature search produced 14,421 papers, 8596 after de-duplication. Based on the title and abstract, 8451 articles were excluded, and 145 full-text articles were thus assessed for eligibility. After this assessment, 27 articles were included for further analysis. Figure 1 shows a flowchart of the selection process. Interrater reliability was considered moderate to good (raw interrater agreement 97.6%; kappa 0.48, 95% CI 0.41–0.55) [11].

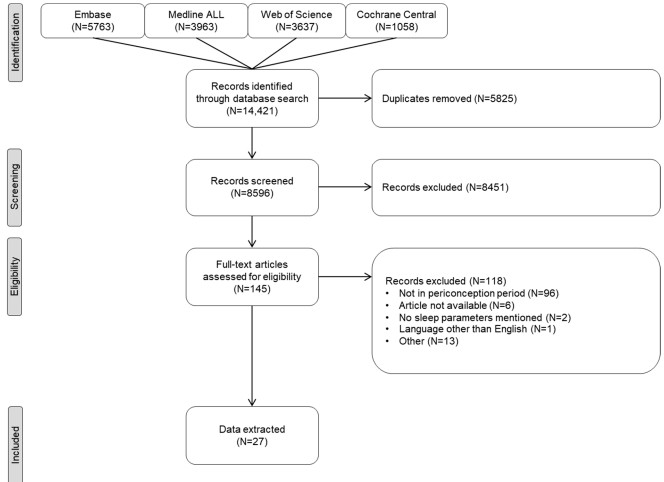

**Figure 1.** PRISMA flow diagram of the article selection process in a systematic review of maternal sleep problems in the periconceptional period. Other reasons included: case-control study (5), sleep was not studied as exposure (3), letter (1), no association with sleep was studied (4).

### 2.2. Study Characteristics

In total, the studies included data on 229,067 participants from 12 countries. However, since a number of studies analyzed the same population, a total of 148,096 unique participants were studied for 12 different outcomes. The sample size per study ranged from 71 to 81,821 participants, with a median of 1116. Of the 27 studies, 20 studies reported maternal outcomes and seven studies reported offspring outcomes. One study studied two different cohorts [12]. Therefore, this systematic review included 28 study samples in total. Of these, 22 were a prospective cohort, three were a retrospective cohort, and three were of cross-sectional design. Detailed study characteristics, including quality scores, are provided in Table 1.

### 2.3. Maternal Outcomes

In this review, 20 studies reported maternal outcomes associated with maternal sleep problems in the periconceptional period (Table 2 and Table S1): three studies investigated fertility, five studies investigated hypertensive disorders, three studies investigated gestational diabetes, seven studies investigated mood, two studies investigated nutrition and weight, and three investigated other outcomes.

**Table 1.** Overview of characteristics of included studies in the systematic review (N = 27).

| Study | Country | Population | Study Design | Sample Size | Exposure Definition | Outcome | Quality Score |
|---|---|---|---|---|---|---|---|
| Bublitz et al. (2021) [13] | USA | Convenience sample of pregnant participants at high risk for SDB at 12 weeks gestation | Prospective cohort | 175 | Objective and self-reported measures of sleep-disordered breathing | Depressive symptoms | 8 |
| Chang et al. (2015) [14] | USA | Low-income and overweight/obese women in their first trimester | Cross-sectional | 75 | PSQI | Depression, stress, fat intake, fruit and vegetable intake | 4 |
| Doyon et al. (2020) [15] | Canada | Women attending a university hospital in their first trimester | Prospective cohort | 766 | Self-reported amount of hours of sleep per night | Physical activity, sedentary behavior, and glycemic regulation | 6 |
| Facco et al. (2018) [a] [16] | USA | Nulliparous women recruited at various clinical sites in their first trimester | Prospective cohort | 7524 | Self-reported sleep times and sleep latency | Gestational hypertensive disorders and gestational diabetes | 8 |
| Facco et al. (2019) [a] [17] | USA | Nulliparous women recruited in various clinical sites in their first trimester | Prospective cohort | 7524 | Self-reported sleep times and sleep latency | Preterm birth | 8 |
| Franco-Sena et al. (2018) [18] | Brazil | Women attending a public health center in their first trimester | Prospective cohort | 176 (65 nulliparous) | Self-reported sleep times | Birth weight | 7 |
| Gelaye et al. (2015) [19] | Peru | Women attending prenatal care clinics in their first trimester | Cross-sectional | 641 | PSQI | Antepartum depression and suicidal ideation | 8 |
| Georgiou et al. (2019) [20] | Greece | Women attending a hospital in their first trimester, retrospectively reporting the presence of sleep disorders before pregnancy | Retrospective cohort | 71 (49 women with problem-free pregnancies and 22 women with preeclampsia) | PSQI, AIS, ESS | Preeclampsia | 3 |
| Haney et al. (2014) [b] [21] | USA | Women recruited in their first trimester by self-referral, physician referral, local advertising, or via University registries | Prospective cohort | 161 | Actigraphy and diary-assessed sleep times | Blood pressure and BMI | 7 |

**Table 1.** *Cont.*

| Study | Country | Population | Study Design | Sample Size | Exposure Definition | Outcome | Quality Score |
|---|---|---|---|---|---|---|---|
| Hill et al. (2021) [22] | USA | Women attending a clinic in their first trimester | Prospective cohort | 339 | PSQI | Gestational weight gain and gestational fat gain | 6 |
| Liu et al. (2019) [23] | China | Women attending a hospital in their first trimester | Prospective cohort | 1466 | PSQI | Birth weight | 9 |
| Lyu et al. (2020) [24] | China | Mothers of children in preschool, retrospectively reporting the amount of hours in their first trimester | Retrospective cohort | 6236 | Self-reported amount of hours of sleep on average per night | Childhood sleep duration and disturbances | 7 |
| Marinelli et al. (2021) [25] | Spain | Women from the general population before conception | Prospective cohort | 2375 | Self-reported sleep duration | Birth weight | 8 |
| Matsuo et al. (2021) [26] | Japan | Women with low-risk pregnancies in their first trimester | Retrospective cohort | 15,314 | Self-reported sleep duration | Postpartum depression | 8 |
| Nakahara et al. (2020) [c] [27] | Japan | Pregnant women visiting a clinic retrospectively reporting on preconception sleep | Prospective cohort | 81,821 | Self-reported sleep duration and bedtime | Preterm birth, offspring sleep, and temperament at 1 month of age | 7 |
| Nakahara et al. (2021) [c] [28] | Japan | Pregnant women visiting a clinic retrospectively reporting on preconception sleep | Prospective cohort | 73,827 | Self-reported sleep duration and bedtime | Offspring sleep and development at 1 year of age | 8 |
| Okada et al. (2019) [29] | Japan | Primipari who visited an obstetrics and gynecology clinic for medical examinations in their first trimester | Prospective cohort | 89 | PSQI | Blood pressure | 5 |
| Okun et al. (2007) [30] | USA | Women reached by paper and e-mail advertisements through the university in their first trimester | Prospective cohort | 78 (35 pregnant; 43 nonpregnant) | PSQI | Cytokine levels | 6 |

**Table 1.** *Cont.*

| Study | Country | Population | Study Design | Sample Size | Exposure Definition | Outcome | Quality Score |
|---|---|---|---|---|---|---|---|
| Okun et al. (2013) [b] [31] | USA | Women recruited in their first trimester by self-referral, physician referral, local advertising, or via University registries | Prospective cohort | 160 | Actigraphy and diary-assessed sleep times | Depressive symptoms and perceived stress | 7 |
| Rawal et al. (2017) [32] | USA | Women in their first trimester participating in a cohort study aimed at studying embryonic growth in various racial groups | Prospective cohort | 2581 (2334 non-obese and 468 obese women) | Self-reported sleep duration | Gestational diabetes | 8 |
| Sarberg et al. (2014) [33] | Sweden | Women visiting an antenatal clinic during their first trimester | Prospective cohort | 500 | Self-reported frequency of snoring | Adverse pregnancy outcomes | 6 |
| Shi et al. (2020) [12] | USA and China | USA: Women from the general population in the preconception period China: Women from the general population in the preconception period | USA: Cross-sectional China: Prospective cohort | USA: 9137 China: 4759 | Self-reported sleep duration | Probability of conception | 7 |
| Stocker et al. (2020) [34] | UK | Three groups in the preconception period: 1. Women with recurrent implantation failure (RIF) attending gynecology outpatient clinics at a tertiary university hospital 2. Women with recurrent miscarriage (RM; three or more unexplained pregnancy losses before < 24 completed weeks of pregnancy), attending gynecology outpatient clinics at a tertiary university hospital 3. Control group consisting of randomly selected gynecology outpatients | Prospective cohort | 88 (34 control, 21 RIF, 33 RM) | PSQI, ESS, actigraphy | Reproductive outcomes | 5 |

**Table 1.** *Cont.*

| Study | Country | Population | Study Design | Sample Size | Exposure Definition | Outcome | Quality Score |
|---|---|---|---|---|---|---|---|
| Tsai et al. (2016) [35] | Taiwan | Women attending a university-affiliated hospital in their first trimester | Prospective cohort | 164 | PSQI, actigraphy | Health-related quality of life | 7 |
| Willis et al. (2019) [36] | USA and Canada | Women from the general population in the preconception period | Prospective cohort | 6873 | Self-reported sleep duration and sleeping problems | Chance of conception | 8 |
| Yu et al. (2017) [37] | China | Women attending a hospital in their first trimester | Prospective cohort | 3645 | Self-reported sleep duration and quality | Depression and anxiety | 7 |
| Yun et al. (2021) [38] | Korea | Women attending a hospital for their pregnancy | Prospective cohort | 2512 | Self-reported sleep duration and quality | Postpartum depression | 7 |

[a] [16,17] analyzed the same study population; [b] [21,31] studied the same study population; [c] [27,28] analyzed the same study population. AIS: Athens Insomnia Scale; BMI: body mass index; ESS: Epworth Sleepiness Scale; PSQI: Pittsburgh Sleep Quality Index.

**Table 2.** Overview of maternal outcomes of included studies in the systematic review (N = 20).

| Study | Exposure Definition | Outcome Definition | Outcome | Quality Score |
|---|---|---|---|---|
| | | Hypertensive disorders | | |
| Facco et al. (2018) [16] | Self-reported first trimester short sleep duration (<7 h) | Hypertensive disorders | OR 1.31 (95% CI 1.10–1.55); $p = 0.002$ * <br> aOR [b] 1.19 (95% CI 1.00–1.42); $p = 0.054$ <br> aOR [c] 1.19 (95% CI 1.00–1.42); $p = 0.053$ | 8 |
| | First trimester late sleep midpoint (after 5 A.M.) | Hypertensive disorders | OR 1.22 (95% CI 1.00–1.49); $p = 0.055$ <br> aOR [b] 1.10 (95% CI 0.89–1.37); $p = 0.367$ <br> aOR [c] 1.15; 95% CI 0.92–1.43; $p = 0.216$ | |
| Georgiou et al. (2019) [20] | Insomnia before pregnancy | Preeclampsia | OR 5.03 (95% CI 1.41–17.89); $p < 0.05$ * | 3 |
| | Sleep quality before pregnancy | Preeclampsia | OR 4.45 (95% CI 1.53–12.99); $p < 0.05$ * | |
| | Sleepiness before pregnancy | Preeclampsia | OR 3.27 (95% CI 1.15–9.31); $p < 0.05$ * | |

**Table 2.** *Cont.*

| Study | Exposure Definition | Outcome Definition | Outcome | Quality Score |
|---|---|---|---|---|
| Haney et al. (2014) [21] | Diary-assessed sleep latency at 10–12 weeks | Systolic blood pressure at 14–16 weeks | r(132) = 0.18, $p$ = 0.03 * Not statistically significant after correcting for multiple comparisons. | 7 |
| | Diary-assessed wake after sleep onset and total sleep time | Any cardiometabolic factor | No statistically significant association. | |
| | Actigraphy-assessed wake after sleep onset and total sleep time | Any cardiometabolic factor | No statistically significant association. | |
| Okada et al. (2019) [29] | First trimester PSQI score | Change in morning systolic blood pressure from first to third trimester | r = 0.49, β = 0.58, $p$ = 0.00 * | 5 |
| | First trimester PSQI subscale sleep latency | Change in morning systolic blood pressure from first to third trimester | r = 0.38, β = 0.43, $p$ = 0.02 * | |
| | First trimester PSQI subscale sleep disturbances | Change in morning systolic blood pressure from first to third trimester | r = 0.24, β = 0.33, $p$ = 0.04 * | |
| | First trimester PSQI subscale subjective sleep quality | Change in morning systolic blood pressure from first to third trimester | r = 0.33; β = 0.30; $p$ = 0.06 | |
| | First trimester PSQI subscale sleep duration | Change in morning systolic blood pressure from first to third trimester | r = 0.26; β = 0.36; $p$ = 0.06 | |
| | First trimester PSQI subscale sleep efficiency | Change in morning systolic blood pressure from first to third trimester | r = 0.37; β = 0.15; $p$ = 0.39 | |
| Sarberg et al. (2014) [33] | First trimester snoring | Systolic blood pressure | $p$ = 0.779 | 6 |
| | | Diastolic blood pressure | $p$ = 0.053 | |
| **Gestational diabetes** | | | | |
| Doyon et al. (2020) [15] | Self-reported first trimester sleep duration | Blood glucose levels at 1 h post 50 g | β = 0.013; SE = 0.007; $p$ = 0.06 aβ [a] = 0.009; SE = 0.007; $p$ = 0.17 | 6 |

**Table 2.** *Cont.*

| Study | Exposure Definition | Outcome Definition | Outcome | Quality Score |
|---|---|---|---|---|
| Facco et al. (2018) [16] | Self-reported first trimester short sleep duration (<7 h) | Gestational diabetes | OR 1.45 (95% CI 1.10–1.92); $p = 0.009$ *<br>aOR [b] 1.26 (95% CI 0.94–1.67); $p = 0.119$<br>aOR [c] 1.23 (95% CI 0.92–1.64); $p = 0.164$ | 8 |
| | First trimester late sleep midpoint (after 5 A.M.) | Gestational diabetes | OR 1.31; 95% CI 0.94–1.82; $p = 0.111$<br>aOR [b] 1.67; 95% CI 1.17–2.38; $p = 0.004$ *<br>aOR [c] 1.37; 95% CI 0.95–1.98; $p = 0.089$ | |
| Rawal et al. (2017) [32] | First trimester sleep duration | Gestational diabetes | No statistical significant association. See Table S1 for more details. | 8 |
| Mood | | | | |
| Bublitz et al. (2021) [13] | Objective measure of sleep-disordered breathing at 12 weeks | Depressive symptoms at 12 weeks | $\beta = 0.11$; $p = 0.16$ | 8 |
| | Self-reported measure of sleep-disordered breathing at 12 weeks | Depressive symptoms at 12 weeks | $F = 0.74$; $p = 0.39$ | |
| | Objective measure of sleep-disordered breathing at 12 weeks | Depressive symptoms at 32 weeks | $\beta = 0.20$; SE = 1.89; $p = 0.026$ *<br>a$\beta$ [d] = 0.22; SE = 1.89; $p = 0.012$ *<br>a$\beta$ [d] = 0.25; SE = 1.60; $p = 0.004$ *<br>(without sleep item)<br>a$\beta$ [d] = 0.19; SE = 1.99; $p = 0.043$ *<br>(without antidepressants) | |
| | Self-reported measure of sleep-disordered breathing at 12 weeks | Depressive symptoms at 32 weeks | $F = 0.43$; $p = 0.51$ | |
| Chang et al. (2015) [14] | First trimester PSQI subscale sleep latency | Depression | effect size 1.27; $p < 0.05$ * | 4 |
| Gelaye et al. (2015) [19] | First trimester PSQI score >5 | Suicidal ideation | OR 2.72 (95% CI 1.78–4.16) *<br>aOR [e] 2.19 (95% CI 1.40–3.42) *<br>aOR [f] 1.67 (95% CI 1.02–2.71) * | 8 |
| | First trimester PSQI score (continuous) | Suicidal ideation | OR 1.26 (95% CI 1.17–1.36) *<br>aOR [e] 1.22 (95% CI 1.13–1.32) *<br>aOR [f] 1.18 (95% CI 1.08–1.28) * | |

**Table 2.** *Cont.*

| Study | Exposure Definition | Outcome Definition | Outcome | Quality Score |
|---|---|---|---|---|
| Matsuo et al. (2021) [26] | First trimester self-reported sleep duration | Postpartum depression | <6 h:<br>- OR 2.06 (95% CI 1.64–2.59) *<br>- aOR [g] 2.08 (95% CI 1.60–2.70) *<br>6–7 h:<br>- OR 1.42 (95% CI 1.21–1.66) *<br>- aOR [g] 1.41 (95% CI 1.18–1.68) *<br>1 h increase:<br>- OR 0.88 (95% CI 0.83–0.94) *<br>- aOR [g] 0.86 (95% CI 0.80–0.92) *<br>See Table S1 for more details. | 8 |
| Okun et al. (2013) [31] | Diary-defined sleep deficiency group (none, mixed, deficiency) at 10–12 weeks | Pregnancy distress | Did not differ between groups. | 7 |
| | | Depressive symptoms | $F_{2,157} = 4.27$; $p = 0.01$, but no longer after adjustment [f]. | |
| | | Stress | $F_{2,157} = 3.51$; $p = 0.03$, but no longer after adjustment [f]. | |
| | Actigraphy-defined sleep deficiency group (none, mixed, deficiency) at 10–12 weeks | Pregnancy distress | $F_{2,157} = 3.96$; $p = 0.02$ * | |
| | | Depressive symptoms | Did not differ between groups. | |
| | | Stress | $F_{2,157} = 6.36$; $p < 0.01$ *<br>Adjusted [h]: $F_{2,152} = 4.57$, $p = 0.01$ * | |
| Yu et al. (2017) [37] | First trimester sleep duration | Depression | $\beta = -0.28$; SE = 0.08; $p < 0.01$ * | 7 |
| | First trimester < 8 h sleep/day | Depression | OR 1.75; 95% CI 1.39–2.20 * | |
| | First trimester fair sleep quality | Depression | OR 1.57 (95% CI 1.34–1.84) * | |
| | First trimester bad sleep quality | Depression | OR 3.27 (95% CI 2.28–4.32) * | |
| | First trimester sleep duration | Anxiety | $\beta = -0.33$; SE = 0.07; $p < 0.01$ * | |
| | First trimester < 8 h sleep/day | Anxiety | OR 2.00, 95% CI 1.57–2.55 * | |
| | First trimester fair sleep quality | Anxiety | OR 2.52 (95% CI 2.06–3.09) * | |
| | First trimester bad sleep quality | Anxiety | OR 7.39 (95% CI 5.89–10.67) * | |

Table 2. *Cont.*

| Study | Exposure Definition | Outcome Definition | Outcome | Quality Score |
|---|---|---|---|---|
| Yun et al. (2021) [38] | Self-reported sleep duration and quality before pregnancy | Postpartum depression | OR 1.37 (95% CI 1.07–1.75); $p = 0.013$ * Not significant after adjustment. | 7 |
| | Self-reported sleep duration and quality at 12 weeks of gestation | Postpartum depression | OR 1.43 (95% CI 1.11–1.83); $p = 0.005$ * Not significant after adjustment. | |
| | | Fertility | | |
| Shi et al. (2020) [12] | Self-reported sleep duration | Conception probability | ≤5 h (USA data): - RR 3.25 (95% CI 2.33–4.53) * - aRR [i] 3.49 (95% CI 2.48–4.91) * - aRR [j] 3.24 (95% CI 2.30–4.58) * 6 h (USA data): - RR 2.04 (95% CI 1.51–2.75) * - aRR [i] 2.17 (95% CI 1.60–2.95) * - aRR [j] 2.11 (95% CI 1.55–2.86) * See Table S1 for more details. | 7 |
| Stocker et al. (2021) [34] | PSQI | Reproductive outcomes | No statistically significant differences between the groups. See Table S1 for more details. | 5 |
| | ESS | Reproductive outcomes | No statistically significant differences between the groups. See Table S1 for more details. | |
| | Diary-assessed sleep parameters | Reproductive outcomes | No statistically significant differences between the groups. See Table S1 for more details. | |
| | Actigraphy-assessed sleep parameters | Reproductive outcomes | Sleep duration (RIF vs. control) $p = 0.03$ * All other sleep parameters were statistically insignificant. See Table S1 for more details. | |
| Willis et al. (2019) [36] | Self-reported sleep duration before pregnancy | Fecundibility | No statistically significant associations. See Table S1 for more details. | 8 |
| | Self-reported sleep problems before pregnancy | Fecundibility | <50% of time: - FR 0.91 (95% CI 0.86–0.97) - aFR [k] 0.93 (95% CI 0.88–1.00) >50% of time: - FR 0.80 (95% CI 0.73–0.87) * - aFR [k] 0.87 (95% CI 0.79–0.95) * See Table S1 for more details. | |

**Table 2.** *Cont.*

| Study | Exposure Definition | Outcome Definition | Outcome | Quality Score |
|---|---|---|---|---|
| Nutrition and weight | | | | |
| Chang et al. (2015) [14] | First trimester PSQI subscale sleep latency | Fruit and vegetable intake | effect size 2.17; $p < 0.05$ * | 4 |
| Hill et al. (2021) [22] | First trimester PSQI score | Inadequate gestational weight gain | OR 1.00 (95% CI 0.88–1.13); $p = 0.99$ | 6 |
| | | Excessive gestational weight gain | OR 0.95 (95% CI 0.86–1.06); $p = 0.35$ | |
| | | Gestational fat gain | $\beta = 0.03$; SE = 0.07; $p = 0.66$ | |
| | First trimester sleep duration | Inadequate gestational weight gain | OR 1.03 (95% CI 0.73–1.46); $p = 0.85$ | |
| | | Excessive gestational weight gain | OR 1.09 (95% CI 0.84–1.42; $p = 0.50$) | |
| | | Gestational fat gain | $\beta = 0.01$; SE = 0.17; $p = 0.97$ | |
| Other | | | | |
| Okun et al. (2007) [30] | First trimester PSQI subscale subjective sleep quality | TNF-$\alpha$ levels | $\rho = 0.41$, $p = 0.02$ * | 6 |
| | | All other biomarkers (IL-4, IL-6, IL-10, CRP) | No statistical significant association. | |
| Sarberg et al. (2014) [33] | First trimester snoring | Restless leg syndrome | $p = 0.147$ | 6 |
| | | Sleepiness | $p = 0.009$ (non-snorers vs. gestational snorers) * $p = 0.264$ (habitual snorers vs. gestational snorers) | |
| Tsai et al. (2016) [35] | First trimester daytime sleep (actigraphy-assessed) | First trimester health-related quality of life | $\beta = 0.03$; $p = 0.04$ * (physical) $\beta = 0.02$; $p = 0.46$ (mental) | 7 |
| | | Second trimester health-related quality of life | $\beta = 0.01$; $p = 0.55$ (physical) $\beta = -0.01$; $p = 0.36$ (mental) | |
| | | Third trimester health-related quality of life | $\beta = -0.01$; $p = 0.59$ (physical) $\beta = 0.01$; $p = 0.43$ (mental) | |
| | First trimester PSQI total score | First trimester health-related quality of life | $\beta = -1.07$; $p < 0.01$ * (physical) $\beta = -1.40$; $p < 0.01$ * (mental) | |
| | | Second trimester health-related quality of life | $\beta = -0.87$; $p < 0.01$ * (physical) $\beta = -0.85$; $p < 0.01$ * (mental) | |
| | | Third trimester health-related quality of life | $\beta = -0.20$; $p = 0.41$ (physical) $\beta = -1.00$; $p < 0.01$ * (mental) | |

**Table 2.** *Cont.*

| Study | Exposure Definition | Outcome Definition | Outcome | Quality Score |
|---|---|---|---|---|
| | First trimester sleep efficiency (actigraphy-assessed) | First trimester health-related quality of life | $\beta = -0.01$; $p = 0.93$ (physical)<br>$\beta = -0.01$; $p = 0.95$ (mental) | |
| | | Second trimester health-related quality of life | $\beta = -0.10$; $p = 0.47$ (physical)<br>$\beta = -0.09$; $p = 0.52$ (mental) | |
| | | Third trimester health-related quality of life | $\beta = -0.27$; $p = 0.06$ (physical)<br>$\beta = 0.07$; $p = 0.62$ (mental) | |
| | First trimester wake after sleep onset (actigraphy-assessed) | First trimester health-related quality of life | $\beta = 0.01$; $p = 0.85$ (physical)<br>$\beta = 0.01$; $p = 0.83$ (mental) | |
| | | Second trimester health-related quality of life | $\beta = -0.02$; $p = 0.56$ (physical)<br>$\beta = -0.05$; $p = 0.12$ (mental) | |
| | | Third trimester health-related quality of life | $\beta = -0.07$; $p = 0.05$ * (physical)<br>$\beta = -0.03$; $p = 0.39$ (mental) | |
| | First trimester total nighttime sleep (actigraphy-assessed) | First trimester health-related quality of life | $\beta = -0.81$; $p = 0.30$ (physical)<br>$\beta = 0.46$; $p = 0.64$ (mental) | |
| | | Second trimester health-related quality of life | $\beta = -0.12$; $p = 0.87$ (physical)<br>$\beta = 1.38$; $p = 0.01$ * (mental) | |
| | | Third trimester health-related quality of life | $\beta = -0.13$; $p = 0.86$ (physical)<br>$\beta = 1.53$; $p = 0.02$ * (mental) | |

* $p < 0.05$. [a] Adjusted for maternal age, pre-pregnancy BMI, gravid status, and smoking status. [b] Adjusted for age and BMI. [c] Adjusted for age, BMI, race/ethnicity, employment status, and insurance status. [d] Adjusted for maternal age, BMI assessed in pregnancy, race, ethnicity, daytime sleepiness (measured by the Epworth Sleepiness Scale in early pregnancy), history of a self-reported depression diagnosis, and self-reported antidepressant medication use in pregnancy. [e] Adjusted for age (years), parity (nulliparous vs. multiparous), access to basics (hard vs. not very hard), and lifetime intimate partner violence (any physical or sexual abuse vs. no abuse). [f] Further adjusted for depression status. [g] Adjusted for maternal age, parity, pre-pregnancy BMI, pregnancy weight gain, history of spontaneous abortion or stillbirth, history of artificial abortion, infertility treatment, smoking status, marital status, duration of education, hypertensive disorders of pregnancy, placental abruption, delivery mode, labor induction, and blood loss at delivery. [h] Adjusted for race/ethnicity, marital status, education, and parity. [i] Adjusted for age, race, marital status, and working status. [j] Further adjusted for BMI, smoking status, alcohol drinking status, vigorous activity, and sleep disorder. [k] Adjusted for female age, BMI, income, non-Hispanic white, prior birth, prior form of hormonal birth control, current smoker, hours working, history of infertility, unemployment, and caffeine consumption. AIS = Athens Insomnia Scale; BMI = body mass index; ESS = Epworth Sleepiness Scale; PSQI = Pittsburgh Sleep Quality Index.

### 2.3.1. Fertility

Three studies investigated the association between sleep and fertility [12,34,36]. Shi et al. studied conception probability in two large cohorts, and found that shorter self-reported sleep duration was associated with a higher conception probability in their cross-sectional cohort, but not in their longitudinal cohort [12]. In both cohorts, a U-shaped association was found and departure from 7 h sleep per day was associated with a higher chance of conception, whether this was shorter or longer. Willis et al., who also studied a large longitudinal cohort, found that more sleep problems were associated with a lower chance of conception [36]. Stocker et al. only found shorter actigraphy-assessed sleep duration to be associated with recurrent implantation failure, among many (objective and subjective) sleep parameters [34].

### 2.3.2. Hypertensive Disorders

Five studies investigated the effects of periconceptional sleep problems on hypertensive disorders [16,20,21,29,33]. One cross-sectional study did not find any statistically significant association [33], whereas longitudinal studies found that poor sleep was associated with more adverse outcomes, although not consistently [16,20,21,29]. Haney et al. found that diary-assessed sleep latency at 10–12 weeks was associated with systolic blood pressure four weeks later, but this was not statistically significant after correcting for multiple comparisons [21]. Other diary-assessed and actigraphy-assessed parameters did not show any statistically significant associations. Okada et al. showed various longitudinal associations between first trimester sleep and changes in morning systolic blood pressure [29]. However, the quality score was only 5, due to the small study sample (N = 89), not using a proper outcome assessment, and not adjusting for additional confounders. Georgiou et al. found that poor sleep before pregnancy strongly predicted the risk of preeclampsia, with ORs ranging from 3.27 to 5.03 [20]. However, the confidence intervals were quite broad (e.g., 1.41–17.89 for insomnia), and the quality score was only 3, due to the small sample size (N = 71), the lack of proper adjustment for confounders, and the high risk for recall bias. Facco et al. showed a statistically significant association between self-reported first trimester short sleep duration, but not late sleep midpoint, and hypertensive disorders [16]. After adjustment for confounders, this association was no longer statistically significant.

### 2.3.3. Gestational Diabetes

In a cross-sectional study, no statistically significant association was found between first trimester self-reported sleep and blood glucose levels after a glucose tolerance test [15]. Two longitudinal studies examined the association between first trimester sleep and gestational diabetes. One study did not find any statistically significant association [32], whereas Facco et al. found that shorter sleep duration and later sleep midpoint were both associated with a higher risk of gestational diabetes, but these associations did not remain significant after adjustment [16].

### 2.3.4. Mood

From all outcomes, associations with mood were most consistent. All studies found that periconceptional sleep problems were associated with stress, depression, anxiety, and/or suicidal ideation [13,14,19,26,31,37,38]. Chang et al. showed that longer sleep latency was associated with a higher risk of depression in the first trimester [14]. Okun et al. did not show statistically significant associations through all analyses consistently but did show that actigraphy-defined sleep deficiency was associated with more perceived stress [31]. Yun et al. found a higher risk of postpartum depression when women reported shorter sleep duration, both before pregnancy and during the first trimester, but this did not remain statistically significant after adjustment [38]. Bublitz et al. showed that objective measures, but not subjective measures, of sleep-disordered breathing at 12 weeks were associated with depressive symptoms at 32 weeks of pregnancy [13]. The studies by Gelaye et al., Yu et al., and Matsuo et al. show considerable effect sizes throughout

all analyses. Gelaye et al. reported ORs ranging from 1.18 to 2.72 for the associations between poor sleep and a higher risk of suicidal ideation in the first trimester [19]. Yu et al. studied the association between poor sleep and a higher risk of depression/anxiety in the first trimester and reported ORs ranging from 1.57 to 7.39 [37]. Matsuo et al. found that shorter self-reported sleep duration in the first trimester was associated with a higher risk of postpartum depression (OR 2.08 for <6 h and OR 1.41 for 6–7 h, after adjustment) [26].

### 2.3.5. Nutrition and Weight

One study found that women with longer first trimester sleep latency were more likely to eat more fruit and vegetables [14]. Hill et al. did not find any statistically significant associations between first trimester sleep and gestational weight gain [22].

### 2.3.6. Other

Okun et al. found that poor first trimester sleep was associated with higher TNF-$\alpha$ levels [30]. No associations were found with all other biomarkers. Sarberg et al. found no statistically significant association between first trimester snoring and restless leg syndrome [33]. They did find that women who started to snore during pregnancy suffered more from sleepiness than non-snorers. Tsai et al. studied the association between first trimester sleep parameters and health-related quality of life in all trimesters [35]. Various associations were found, indicating that more sleep problems are associated with lower quality of life, but these were not consistent throughout all analyses. PSQI-assessed outcomes were more consistent compared to actigraphy-assessed outcomes.

### 2.4. Offspring Outcomes

In this review, seven studies reported offspring outcomes associated with maternal sleep in the periconceptional period (Table 3 and Table S1): two studies investigated preterm birth, three studied birth weight, and three studied offspring sleep and behavior.

**Table 3.** Overview of offspring outcomes of included studies in the systematic review (N = 5).

| Study | Exposure Description | Outcome Definition | Outcome | Quality Score |
|---|---|---|---|---|
| | | Preterm birth | | |
| Facco et al. (2019) [17] | Self-reported sleep duration (<7 h) in the first trimester | All preterm birth | OR 1.17 (95% CI 0.94–1.47); $p = 0.16$<br>aOR [a] 1.15 (95% CI 0.92–1.44); $p = 0.23$ | 8 |
| | | Spontaneous preterm birth | OR 1.11 (95% CI 0.84–1.46); $p = 0.48$<br>aOR 1.11 (95% CI 0.84–1.47); $p = 0.47$ | |
| | Late sleep midpoint (after 5 A.M.) in the first trimester | All preterm birth | OR 1.42 (95% CI 1.11–1.82); $p = 0.005$ *<br>aOR [a] 1.39 (95% CI 1.08–1.80); $p = 0.01$ *<br>aOR [b] 1.34 (95% CI 1.03–1.74); $p = 0.03$ * | |
| | | Spontaneous preterm birth | OR 1.43 (95% CI 1.06–1.93); $p = 0.019$ *<br>aOR [a] 1.45 (95% CI 1.06–1.99); $p = 0.02$ *<br>aOR [b] 1.34 (95% CI 0.97–1.85); $p = 0.07$ | |
| Nakahara et al. (2020) [27] | Pre-pregnancy self-reported sleep duration | Preterm birth | No statistically significant associations. See Table S1 for more details. | 7 |
| | Pre-pregnancy self-reported bedtime | Preterm birth | No statistically significant associations. See Table S1 for more details. | |
| | | Birth weight | | |
| Franco-Sena et al. (2018) [18] | First trimester: 24 h sleep duration nulliparous | Birth weight | β = −0.35 (95% CI 0.57–0.14); $p = 0.07$ *<br>aβ [c] = 0.44 (95% CI 0.68–0.21); $p < 0.001$ *<br>aβ [d] = −0.42 (95% CI 0.65–0.18); $p < 0.001$ * | 7 |
| | First trimester: nightly sleep duration nulliparous | Birth weight | β = −0.28 (95% CI 0.53–0.03); $p = 0.029$ *<br>aβ [c] = −0.27 (95% CI 0.58–0.05); $p = 0.092$ | |
| | First trimester: napping sleep duration nulliparous | Birth weight | β = −0.08 (95% CI 0.60–0.44); $p = 0.757$<br>aβ [c] = −0.20 (95% CI 0.77–0.37); $p = 0.485$ | |
| | First trimester: 24 h sleep duration multiparous | Birth weight | β = −0.09 (95% CI 0.25–0.07); $p = 0.277$<br>aβ [c] = −0.36 (95% CI 1.39–0.66); $p = 0.483$ | |
| | First trimester: nightly sleep duration multiparous | Birth weight | cβ = −0.07 (95% CI 0.28–0.13); $p = 0.478$<br>aβ [c] = −0.01 (95% CI 0.24–0.22); $p = 0.954$ | |
| | First trimester: napping sleep duration multiparous | Birth weight | β = −0.13 (95% CI 0.62–0.35); $p = 0.580$<br>aβ [c] = 0.11 (95% CI 0.49–0.71); $p = 0.719$ | |

**Table 3.** *Cont.*

| Study | Exposure Description | Outcome Definition | Outcome | Quality Score |
|---|---|---|---|---|
| Liu et al. (2021) [23] | First trimester PSQI score | Birth weight (female) | r = −0.093; $p < 0.05$ *<br>aβ [e] = −0.029 (95% CI 0.057–0.001); $p = 0.045$ *<br>aβ [f] = −0.032 (95% CI 0.063–0.001), $p = 0.043$ * | 9 |
| | | Birth weight (male) | r = 0.022; $p > 0.05$<br>aβ [e] = 0.027 (95% CI 0.002–0.057); $p = 0.071$<br>aβ [f] = 0.026 (95% CI 0.006–0.058); $p = 0.113$ | |
| | | Small for gestational age | aOR [e] 1.052 (95% CI 0.892–1.241); $p = 0.546$<br>aOR [f] 1.040 (95% CI 0.865–1.250); $p = 0.678$ | |
| | | Low birth weight | aOR [e] 1.010 (95% CI 0.733–1.391); $p = 0.953$<br>aOR [f] 1.170 (95% CI 0.821–1.669); $p = 0.386$ | |
| Marinelli et al. (2021) [25] | Self-reported sleep duration before pregnancy | Birth weight | <7 h: β = 44.72 (95% CI 0.28–89.17); $p = 0.049$ *<br>≥7 and <9 h: β = 15.75 (95% CI −9.37–40.86); $p = 0.219$<br>≥9 h: β = −39.22 (95% CI −61.46–16.97); $p = 0.001$ *<br>See Table S1 for more details. | 8 |
| Offspring sleep and behavior | | | | |
| Lyu et al. (2020) [24] | First trimester sleep duration (<8 h) | Sleep duration (<10 h) | OR 1.24; 95% CI 1.12–1.38; $p < 0.01$ *<br>aOR [g] 1.25; 95% CI 1.12–1.39; $p < 0.01$ * | 7 |
| | | Sleep disturbance | OR 1.28; 95% CI 1.04–1.59; $p < 0.01$ *<br>aOR [g] 1.13; 95% CI 0.90–1.42; $p > 0.05$ | |
| Nakahara et al. (2020) [27] | Pre-pregnancy self-reported sleep duration | Awakenings | No statistically significant associations. See Table S1 for more details. | 7 |
| | | Tendency to sleep longer during the day than the night | <6 h: aRR [h] 1.18 (95% CI 1.12–1.25) *<br>6–7 h: aRR [h] 1.10 (95% CI 1.06–1.15) *<br>See Table S1 for more details. | |
| | | Bad mood | <6 h: aRR [h] 1.12 (95% CI 1.01–1.23) *<br>6–7 h: aRR [h] 1.09 (95% CI 1.01–1.16) *<br>>10 h: aRR [h] 1.17 (95% CI 1.02–1.33) *<br>See Table S1 for more details. | |
| | | Frequent crying | <6 h: aRR [h] 1.17 (95% CI 1.11–1.24) *<br>6–7 h: aRR [h] 1.09 (95% CI 1.05–1.13) *<br>See Table S1 for more details. | |

**Table 3.** *Cont.*

| Study | Exposure Description | Outcome Definition | Outcome | Quality Score |
|---|---|---|---|---|
| | | Intense crying | <6 h: aRR [h] 1.15 (95% CI 1.09–1.20) *<br>6–7 h: aRR [h] 1.08 (95% CI 1.04–1.12) *<br>See Table S1 for more details. | |
| | Pre-pregnancy self-reported bedtime | Awakenings | No statistically significant associations. See Table S1 for more details. | |
| | | Tendency to sleep longer during the day than the night | 24:00–03:00: aRR [h] 1.17 (95% CI 1.13–1.20) *<br>Other: aRR [h] 1.13 (95% CI 1.04–1.22) *<br>See Table S1 for more details. | |
| | | Bad mood | 24:00–03:00: aRR [h] 1.12 (95% CI 1.06–1.19) *<br>See Table S1 for more details. | |
| | | Frequent crying | 24:00–03:00: aRR [h] 1.09 (95% CI 1.06–1.13) *<br>See Table S1 for more details. | |
| | | Intense crying | 24:00–03:00: aRR [h] 1.07 (95% CI 1.04–1.10) *<br>See Table S1 for more details. | |
| Nakahara et al. (2021) [28] | Pre-pregnancy self-reported sleep duration | >3 nighttime waking instances | 9–10 h: aRR [i] 1.20 (95% CI 1.02–1.40) *<br>See Table S1 for more details. | 8 |
| | | >1 waking instance lasting >1 h | <6 h: aRR [i] 1.49 (95% CI 1.34–1.66) *<br>6–7 h: aRR [i] 1.16 (95% CI 1.07–1.26) *<br>>10 h: aRR [i] 1.25 (95% CI 1.09–1.44) *<br>See Table S1 for more details. | |
| | | <8 h of sleep during the night | <6 h: aRR [i] 1.60 (95% CI 1.44–1.79) *<br>6–7 h: aRR [i] 1.19 (95% CI 1.09–1.29) *<br>>10 h: aRR [i] 1.26 (95% CI 1.09–1.46) *<br>See Table S1 for more details. | |
| | | Falling asleep at 22:00 or later | <6 h: aRR [i] 1.33 (95% CI 1.26–1.40) *<br>6–7 h: aRR [i] 1.15 (95% CI 1.10–1.19) *<br>9–10 h: aRR [i] 0.84 (95% CI 0.79–0.89) *<br>See Table S1 for more details. | |

**Table 3.** *Cont.*

| Study | Exposure Description | Outcome Definition | Outcome | Quality Score |
|---|---|---|---|---|
| | | Frequency of crying at night (≥5 days/week) | <6 h: aRR [i] 1.16 (95% CI 1.05–1.29) * See Table S1 for more details. | |
| | | Communication | No statistically significant associations. See Table S1 for more details. | |
| | | Gross motor skills | 6–7 h: aRR [i] 1.11 (95% CI 1.02–1.20) * See Table S1 for more details. | |
| | | Fine motor skills | No statistically significant associations. See Table S1 for more details. | |
| | | Problem-solving | No statistically significant associations. See Table S1 for more details. | |
| | | Personal-social characteristics | 8–9 h: aRR [i] 1.23 (95% CI 1.02–1.48) * 9–10 h: aRR [i] 1.30 (95% CI 1.03–1.65) * See Table S1 for more details. | |
| | | Total (abnormal score for any 1 of the 5 domains) | No statistically significant associations. See Table S1 for more details. | |
| | Pre-pregnancy self-reported | >3 nighttime waking instances | 24:00–03:00: aRR [i] 0.89 (95% CI 0.80–0.99) * See Table S1 for more details. | |
| | | >1 waking instance lasting >1 h | 24:00–03:00: aRR [i] 1.38 (95% CI 1.30–1.47) * Other: aRR [i] 1.92 (1.67–2.21) * See Table S1 for more details. | |
| | | <8 h of sleep during the night | 24:00–03:00: aRR [i] 1.31 (95% CI 1.22–1.40) * Other: aRR [i] 2.04 (95% CI 1.77–2.35) * See Table S1 for more details. | |
| | | Falling asleep at 22:00 or later | 24:00–03:00: aRR [i] 1.53 (95% CI 1.48–1.58) * Other: aRR [i] 1.34 (95% CI 1.23–1.45) * See Table S1 for more details. | |
| | | Frequency of crying at night (≥5 days/week) | No statistically significant associations. See Table S1 for more details. | |

**Table 3.** *Cont.*

| Study | Exposure Description | Outcome Definition | Outcome | Quality Score |
|---|---|---|---|---|
| | | Communication | No statistically significant associations. See Table S1 for more details. | |
| | | Gross motor skills | No statistically significant associations. See Table S1 for more details. | |
| | | Fine motor skills | No statistically significant associations. See Table S1 for more details. | |
| | | Problem-solving | No statistically significant associations. See Table S1 for more details. | |
| | | Personal-social characteristics | No statistically significant associations. See Table S1 for more details. | |
| | | Total (abnormal score for any 1 of the 5 domains) | No statistically significant associations. See Table S1 for more details. | |

* $p < 0.05$; [a] Adjusted for age and BMI; [b] Adjusted for age, BMI, race/ethnicity, employment status, and insurance status; [c] Adjusted for maternal age, pre-pregnancy BMI, smoking status, Edinburgh Postnatal Depressive Scale (EPDS) at baseline, total gestational weight gain adequacy, leisure-time physical activity, education, marital status, per-capita family income, and planned pregnancy; [d] Further adjusted for second- and third-trimester caloric intake; [e] Adjusted for maternal age (years), ethnicity, education level, pre-pregnancy BMI (kg/m$^2$), gestational weight gain (kg), parity, gestational age (days) and antepartum obstetric risk; [f] Further adjusted for trimester-specific psychosocial status including the perception of stress, symptoms of depression and anxiety levels; [g] Adjusted for maternal age at delivery, maternal education level, overweight/obesity before pregnancy, family structure, family income, smoking during pregnancy, alcohol use during pregnancy, physical activity during pregnancy, maternal stressful life events, family history of sleep disorder, children's age, gender, obesity/overweight, allergic disease, ADHD, childhood physical activity, and screen exposure; [h] Adjusted for maternal age at delivery, smoking habits, alcohol consumption, pre-pregnancy body mass index, parity, current history of diabetes or gestational diabetes, hypertensive disorders in pregnancy and intrauterine infection, history of preterm birth, and infertility treatment. [i] Adjusted for maternal age at delivery, smoking habits, alcohol consumption, pre-pregnancy body mass index, gestational age at birth, parity, infertility treatment, and infant sex.

### 2.4.1. Preterm Birth

The relation between periconceptional sleep and preterm birth was studied in two prospective cohorts (N = 7524 and N = 81,821). Sleep duration before pregnancy and in the first trimester did not seem to affect preterm birth [17,27]. However, later midpoint sleep in the first trimester was associated with a higher risk of preterm birth [17].

### 2.4.2. Birth Weight

Three studies assessed the associations between periconceptional sleep and birth weight [18,23,25]. One study found only an association in female offspring, with more sleep problems in the first trimester being associated with lower birth weight [23], and the other study found only an association in nulliparous women, with a greater decrease in first trimester sleep duration resulting in lower birth weight [18]. One study showed that in women sleeping < 7 h per day before pregnancy, each additional hour of sleep increased birth weight by 44.7 g, while in women sleeping > 9 h per day, each additional hour of sleep decreased birth weight by 39.2 g [25].

### 2.4.3. Offspring Sleep and Behavior

Three studies reported the associations between periconceptional sleep problems and offspring sleep and behavior [24,27,28]. All studies, two studying the preconceptional period and one studying the first trimester, found some associations indicating that more maternal sleep problems were associated with more problems with offspring behavior, such as intense and frequent crying, but not for all of their outcome measurements, such as problems with communication.

### 2.5. Quality Assessment

An overview of the quality assessment can be found in Figures S1 and S2. Table 1 shows the quality scores per study. The quality of the studies, according to the ErasmusAGE criteria [39], ranged from 3 to 9. Since intervention studies were excluded from this systematic review, 9 was the highest score possible. In this review, 25 cohorts were of longitudinal design and three cohorts were of cross-sectional design. Considering the study size, five studies examined < 100 participants, seven studied 100–500 participants, and 15 studied > 500 participants. With regards to the method of exposure assessment, two studies used an inappropriate method, 14 studies used a method of moderate quality, and 11 studies a method of adequate quality. Considering the method of outcome assessment, no studies used an inappropriate method, four studies used a method of moderate quality, and 23 studies used a method of adequate quality. For adjustment for confounding factors, six studies did not confound for at least one key confounder, seven studies controlled for key confounders, and 14 studies were additionally controlled for additional confounders.

## 3. Discussion

### 3.1. Main Findings

Three studies investigated the association between sleep and fertility, which drew different conclusions. A recent meta-analysis showed that both female and male fertility, as well as IVF outcomes, may be affected by short sleep duration, later chronotype, and working night shifts [40]. Here, the circadian rhythm is often disturbed, which may be an additional underlying risk factor for subfertility [41]. Circadian rhythm disturbances may cause changes in the hypothalamus-pituitary-adrenal gland (HPA) axis, which does not only affect cortisol levels, but also other hormones involved that may affect reproduction, such as estrogens, androgens, and melatonin [41].

Five studies assessed the effects of periconceptional sleep on blood pressure. Some statistically significant associations were found, but these were not consistent, especially when adjustments, study quality, and corrections were taken into account. A recent meta-analysis studying 120 articles showed that sleep disturbances during pregnancy are associated with a higher risk of preeclampsia and gestational hypertension [42]. However, the focus was

here on the third trimester, where both sleep problems and hypertensive disorders are more prevalent [1]. This larger variation may have provided more statistical power to detect potential associations. Another possibility is that sleep in the third trimester may be of more importance compared to the periconceptional period. Various meta-analyses have also shown similar associations between sleep disturbances and hypertension outside of pregnancy [43,44]. The direction of this association is not entirely clear and therefore, it is difficult to differentiate between cause and effect, and it even may be bidirectional. Although we did not find consistent associations between sleep and hypertension, we hypothesize that the periconceptional period is possibly no exception in the life course, but that choices in study design, exposure, and outcome measurements made it difficult to find potential associations. Similar conclusions may be drawn from studies regarding gestational diabetes. This systematic review did not suggest a statistically significant association with periconceptional sleep problems, but a previous meta-analysis showed that sleep disturbances during pregnancy as a whole are associated with gestational diabetes [42]. Outside pregnancy, meta-analyses have shown that poor sleep quality is a predictor for the development of type 2 diabetes [44,45].

From all outcomes, associations with mood were the most consistent. All studies found that periconceptional sleep problems were associated with stress, depression, anxiety, and/or suicidal ideation. A relationship between sleep and mood has been shown consistently during pregnancy [46–48], postpartum [49–51], and outside of pregnancy [52,53]. The present systematic review shows that the periconceptional period is no exception here.

Two studies examined the effects of periconceptional sleep on the risk of preterm birth. Sleep duration did not seem to affect this, whereas late midpoint sleep in the first trimester did. A late midpoint sleep is indicative of a later chronotype, being a so-called 'night owl', which could in itself be a risk factor for preterm birth [54]. However, a preference towards the evening is also associated with more unhealthy lifestyle habits [54] and a higher risk for depression [55,56], which could both be involved in this association.

Three studies investigated the associations between periconceptional sleep and birth weight. One study found only an association in female offspring, one study found only an association in nulliparous women, and one study found that the relationship between preconceptional sleep and birth weight is a U-shaped curve, where both a lack and an excess of sleep may impact birth weight. To our knowledge, there is no plausible biological explanation for why sleep would only affect female offspring. Moreover, the effect was only small, and this might therefore be a spurious finding and/or not clinically relevant. It is unclear why this association would only be found in nulliparous women and not in multiparous women. Additionally, this association was only found for 24-h sleep duration and not for nightly sleep duration or for napping duration. As such, there is little evidence from these studies for an association between periconceptional sleep and birth weight, outside of findings in specific subgroups which could plausibly be statistical artifacts.

Three studies reported the associations between periconceptional sleep problems and offspring sleep and behavior. These studies found some associations, but not for all of their outcome measurements. It is important to note that both studies by Nakahara et al. studied the same cohort and reported a large number of statistical analyses, therefore, their outcomes must be interpreted with caution.

### 3.2. Strengths and Limitations

A major strength of our study is the systematic approach which has been conducted and reported in line with PRISMA guidelines. Various systematic reviews and meta-analyses have studied the effects of sleep problems during pregnancy, but this is the first to study sleep problems in the periconceptional period specifically. Moreover, we did not only include studies regarding the maternal effects but also included studies reporting offspring outcomes.

In terms of limitations, between-study variability was the biggest challenge. Differences in study design, populations, methods of exposure, and outcome assessment made

it difficult to compare the findings and impossible to conduct a meta-analysis. Another limitation was that, since we focused on the periconceptional period, we do not have information on sleep during the second and third trimesters, which could have potentially impacted these findings. We have no clinical information about the women that might affect sleep and/or any outcomes, such as pharmacological treatment, pre-existing conditions, or environmental conditions. It is possible that not all studies adequately adjusted for this. No studies assessed the whole periconceptional period. Studies only assessed sleep either before pregnancy or during the first trimester. Lastly, only two out of 22 studies had a low risk of bias on the four quality criteria, indicating that the quality of most included studies is suboptimal.

### 3.3. Interpretation

Sleep problems influence multiple pathways, such as those involved in endocrinology, metabolism, and immunology, which might explain the various associations [57]. For example, it has been hypothesized that sleep deprivation may lead to insulin resistance, which in turn may lead to the development of diabetes [58,59]. Additionally, sleep problems are associated with increased cortisol levels, suggesting changes in the HPA axis, which may explain the association with mood problems [60], but also with birth weight [61]. In turn, sleep problems influence various lifestyle behaviors, such as eating, which may be an underlying mechanism in the development of adverse outcomes, such as obesity [62].

The magnitude and the direction of these associations with periconceptional sleep are not entirely clear, as it is difficult to study how cause and effect relate to one another. These various associations are complex, multidimensional, often bidirectional, and impact various pathways. Moreover, the associations are possibly outcome specific, which makes it difficult to speculate on one specific biological mechanism.

## 4. Materials and Methods

This systematic review was registered in PROSPERO under number CRD42021234111. Due to COVID, the acceptance of our registration was delayed and therefore, it took place after our initial search.

### 4.1. Search

A medical information specialist conducted the literature search on 23 November 2020 and updated on 23 September 2021. The search was conducted in Embase, Cochrane Central, MedLine Ovid, and Web of Science from inception onwards. The search consisted of keywords for the periconceptional period and maternal sleep. The full search strategy can be found in the Supplementary Material.

### 4.2. Study Criteria

This systematic review was conducted and reported in line with PRISMA guidelines [63]. We included observational studies that described any maternal sleep problem in the periconceptional period (defined as the time between 14 weeks before conception and 10 weeks after) and associations with maternal and/or offspring outcomes. We did not choose a specific type of sleep problem but included all studies that mentioned any problems with sleep. We excluded RCTs and any other interventional study, case-control studies, systematic reviews, meta-analyses, case reports, case series, and conference abstracts. Studies that reported on maternal sleep problems during pregnancy in general, without specifying when were excluded. Studies studying sleep problems in the first trimester were included since these largely overlap with the periconceptional period. No restrictions were set for the year of publication.

### 4.3. Study Selection and Data Extraction

Duplicates were screened and removed with the citation manager EndNote [64]. Two reviewers (BB, MZ) independently screened the titles and abstracts and subsequently

independently assessed the full texts of eligible studies. Mismatches between reviewers' selections were resolved by discussion until a consensus was reached.

The two reviewers extracted the following data using a data extraction form: title, authors, year of publication, country, study design, study population, method of recruitment, sleep measure (such as actigraphy and questionnaires), parameters measured regarding maternal sleep (such as sleep latency, sleep quality, etc.), effects on the mother and/or offspring, and effect sizes and/or levels of significance.

### 4.4. Quality

The reviewers assessed the quality of the studies using the ErasmusAGE, a tool composed of five items [39]. Each of the five items was allocated either 0, 1, or 2 points, giving a total score from 0 to 10, with higher scores indicating higher quality. The five items included study design (0 = cross-sectional study, 1 = longitudinal study, 2 = intervention study), study size ($0 \leq 100$, 1 = 100–500, $2 \geq 500$), method of exposure assessment (0 = not appropriate or not reported, 1 = moderate quality (e.g., self-report), 2 = good quality (e.g., validated questionnaires or actigraphy)), method of outcome assessment (0 = not appropriate or not reported, 1 = moderate quality (e.g., not validated questionnaires or self-assessed), 2 = good quality (e.g., medical records)), and analysis with adjustment for potential confounders (0 = not controlled for key confounders, 1 = controlled for key confounders (maternal age, BMI, and parity), 2 = additionally controlled for additional confounders, such as marital status and education ). Since we excluded intervention studies, the maximum score possible was 9.

### 4.5. Analysis

Due to the heterogeneity of the outcomes, a meta-analysis was not possible and we therefore qualitatively reviewed the literature. We categorized the studies into maternal and offspring outcomes. In these two categories, we made various subcategories, depending on identified outcomes (e.g., hypertensive disorders, birth weight). The various associations are compared and discussed. We evaluated how sleep problems were defined, how outcomes were defined, and we evaluated the quality of the studies.

## 5. Conclusions

This systematic review shows that periconceptional sleep problems are associated with various adverse outcomes in both mother and offspring, such as a higher risk of maternal hypertensive disorders and of preterm birth, with the association between sleep problems and maternal mood disorders being the most consistent. However, studies are heterogeneous and findings are not consistent, which makes it difficult to draw stringent conclusions. Given the high prevalence, future research should continue to study the short-term and long-term effects of sleep problems in the periconceptional period, preferably in a longitudinal design. In a design such as this, starting from the preconceptional period and including pregnancy and the postpartum period, it would be possible to study whether sleep problems in the preconceptional period have a large effect on both mother and offspring or whether these effects can be overcome by later antepartum and postpartum experiences. Moreover, healthcare providers should pay attention to sleep problems prior to and at least in the first trimester of pregnant patients, and not wait until later in pregnancy.

**Supplementary Materials:** The following supporting information can be downloaded at: https://www.mdpi.com/article/10.3390/clockssleep4040052/s1, Search terms used in the current systematic review; Table S1: Extensive overview of associations of included studies regarding maternal sleep problems in the periconceptional period (N = 27); Figure S1: Risk of bias assessment per study, based on the ErasmusAGE (N = 27); Figure S2: Risk of bias assessment per criterion (based on the ErasmusAGE).

**Author Contributions:** Conceptualization, B.B., L.v.R. and R.P.M.S.-T.; methodology, B.B. and L.v.R.; formal analysis, B.B. and M.G.Z.; investigation, B.B., M.G.Z. and A.I.L.; resources, A.I.L.; data curation, B.B. and M.G.Z.; writing—original draft preparation, B.B. and M.G.Z.; writing—review and editing, all authors; supervision, L.v.R. and R.P.M.S.-T.; project administration, R.P.M.S.-T. All authors have read and agreed to the published version of the manuscript.

**Funding:** This research received no external funding.

**Institutional Review Board Statement:** Not applicable.

**Informed Consent Statement:** Not applicable.

**Data Availability Statement:** Not applicable.

**Acknowledgments:** The authors thank Wichor Bramer, from Erasmus MC Medical Library for developing the search strategy.

**Conflicts of Interest:** The authors declare no conflict of interest.

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
