# Peer review of "Maternal Sleep Problems in the Periconceptional Period and the Impact on Health of Mother and Offspring: A Systematic Review"

_2624-5175, doi:10.3390/clockssleep4040052_

Round 1

Reviewer 1 Report

This manuscript reports a systematic review of the literature on maternal and infant outcomes due to sleep in the periconceptual period. The authors state this is the first review of periconceptual sleep, and included 27 studies. Generally they found that periconceptual sleep was related to lower fertility and higher mood disorders in mothers along with some adverse outcomes in offspring.

This manuscript provides answers to an important question, and with some reorganization and additions, it could be considered for publication in Clocks & Sleep.

Beginning with the abstract, the following changes should be made:

Abstract: should be attenuated in its tone with regard to the results of the systematic review. For example, line 22-23 includes hypertensive disorders as an outcome, but in the text the authors discuss many problems with that research and provide reasons for the reader as to why the results cannot be relied upon from those studies. So that doesn’t seem to be an outcome to be included in the abstract. Similarly, higher risk of preterm birth (line 23) and low birth weight (line 24) is questionable in the findings (as discussed in the text), so those inconsistencies should be noted in the abstract, rather than being presented there as clear findings. Similarly, the abstract notes that offspring behavior is affected, but details of what is affected are not discussed in the text (they are presented in Table S1 but then left up to the reader to find among all the results of those studies. The end of the abstract states that sleep in the periconceptual period is crucial, but this does not seem to follow from the results of the review.

The Materials & Methods section should appear before the results, and the “study selection” section paragraph (currently 2.1) should appear with Figure 1 in the Materials & Methods section.

Some places in the manuscript could benefit from more detail being included:

p. 2, line 48 – what are some of the diseases that affect people later in life?

Figure 1 (PRISMA flowchart), p. 3, records excluded box should include a list of reasons “other (n=13) were excluded – or that list could be in the text of materials & methods

Current section 2.4.3 (offspring sleep & behavior), p. 20, what outcome measurements were found to be associated with sleep in the periconceptual period, and what outcomes were not? And refer the reader specifically to Table SI

In the Quality assessment paragraph (currently section 2.5, p. 20), make sure to discuss the issue of bias and how it was assessed – refer back that Table 1 has the quality scores.

In the discussion, clarify on p. 21, paragraph beginning on line 276 that the conclusion here is that the effects on blood pressure may not be reliable as they are not consistent. Be careful in discussing “our hypothesis” on line 289 – the evidence is not showing that the periconceptual period is important in contributing to hypertension, but the authors still hypothesize it is “no exception to the life course” – but your conclusions need to be evidence-based. If there is no evidence for it, how do you know the periconceptual period is NOT different from later stages of pregnancy? Research is about finding evidence-based conclusions, not promoting foregone conclusions, so be careful here.

On p. 21 there are some places where it reads “xx studies studied…” – the wording is a bit awkward, try “examined” or “tested” or some other verb here than studied.

Mention in current section 4.2 (study criteria) that intervention studies were excluded. This is mentioned later, but should also appear in the exclusion list. This is also where the reasons for exclusion “other (n=13)” from the PRISMA flowchart could be listed if not added to the figure.

P. 23, current section 4.4, line 408 – list some of the additional confounders that some studies adjusted for (as examples).

The conclusion section (p. 23) should include a bit more information (more like the abstract) in specifically what results were found (a brief overview of the general findings). In addition, there is a need for longitudinal studies including the periconceptual period to see if changes in maternal sleep from the periconceptual period to birth result in different maternal or child outcomes. This would clarify whether the periconceptual period was “vital” for later health & development, or whether adversity during this period could be overcome by later prenatal experiences.

Author Response

This manuscript reports a systematic review of the literature on maternal and infant outcomes due to sleep in the periconceptual period. The authors state this is the first review of periconceptual sleep, and included 27 studies. Generally they found that periconceptual sleep was related to lower fertility and higher mood disorders in mothers along with some adverse outcomes in offspring.

This manuscript provides answers to an important question, and with some reorganization and additions, it could be considered for publication in Clocks & Sleep.

- First, we would like to thank reviewer 1 for their time and effort to review the manuscript. It is clear the reviewer engaged extensively and thoughtfully with the study. We appreciate the suggestions for improving our manuscript, and have added the details and changes in wording the reviewer requested.

Beginning with the abstract, the following changes should be made:

Abstract: should be attenuated in its tone with regard to the results of the systematic review. For example, line 22-23 includes hypertensive disorders as an outcome, but in the text the authors discuss many problems with that research and provide reasons for the reader as to why the results cannot be relied upon from those studies. So that doesn’t seem to be an outcome to be included in the abstract. Similarly, higher risk of preterm birth (line 23) and low birth weight (line 24) is questionable in the findings (as discussed in the text), so those inconsistencies should be noted in the abstract, rather than being presented there as clear findings. Similarly, the abstract notes that offspring behavior is affected, but details of what is affected are not discussed in the text (they are presented in Table S1 but then left up to the reader to find among all the results of those studies). The end of the abstract states that sleep in the periconceptual period is crucial, but this does not seem to follow from the results of the review.

- We recognize the comment of the reviewer about attenuating the tone and leaving the reader with Table S1 on their own. We made changes accordingly with which we hope solve the issue. We changed the sentence to: “Some associations were found between sleep problems and lower fertility, more hypertensive disorders, and more mood disorders in mothers, higher risk of preterm birth and lower birth weight, and more sleep and behavior problems in offspring, ...”, to make the wording less strong. We also added “...although not consistently” to line 27.

We understand why reviewer 1 might conclude sleep appears less crucial due to the results of this review. However, this review does show that there is a consistent association between sleep and mood. With approximately 12% of pregnant women suffering from antepartum and postpartum depression, we do believe that sleep is crucial. Therefore, we added “especially for maternal mood” to line 28, to make this conclusion more specific.

The Materials & Methods section should appear before the results, and the “study selection” section paragraph (currently 2.1) should appear with Figure 1 in the Materials & Methods section.

- We are also used to placing Methods before Results like the reviewer suggests. However, the guidelines of the journal state that the results are mentioned after the introduction and that the methods are mentioned at the end of the manuscript. Unfortunately, we cannot change this.

Some places in the manuscript could benefit from more detail being included:

  1. 2, line 48 – what are some of the diseases that affect people later in life?

- Agreed. We added “such as hypertensive disorders and diabetes” to this sentence.

Figure 1 (PRISMA flowchart), p. 3, records excluded box should include a list of reasons “other (n=13) were excluded – or that list could be in the text of materials & methods.

- The specific reasons have now been included in the text below the figure.

Current section 2.4.3 (offspring sleep & behavior), p. 20, what outcome measurements were found to be associated with sleep in the periconceptual period, and what outcomes were not? And refer the reader specifically to Table SI.

- We added which problems were found to be associated with sleep and which not (see section 2.4.3).

In the Quality assessment paragraph (currently section 2.5, p. 20), make sure to discuss the issue of bias and how it was assessed – refer back that Table 1 has the quality scores.

- We added the following sentence: “Table 1 shows the quality scores per study.” In the methods section, section 4.4, we also thoroughly discussed how the quality is assessed per study.

In the discussion, clarify on p. 21, paragraph beginning on line 276 that the conclusion here is that the effects on blood pressure may not be reliable as they are not consistent. Be careful in discussing “our hypothesis” on line 289 – the evidence is not showing that the periconceptual period is important in contributing to hypertension, but the authors still hypothesize it is “no exception to the life course” – but your conclusions need to be evidence-based. If there is no evidence for it, how do you know the periconceptual period is NOT different from later stages of pregnancy? Research is about finding evidence-based conclusions, not promoting foregone conclusions, so be careful here.

- We agree with the reviewer that our wording was too strong and should instead reflect the uncertainty of the empirical literature we reviewed. Therefore, we changed the sentence to “we hypothesize that the periconceptional period is possibly no exception in the life course, ...”

On p. 21 there are some places where it reads “xx studies studied…” – the wording is a bit awkward, try “examined” or “tested” or some other verb here than studied.

- We agree with the reviewer. Throughout our manuscript, we have changed these sentences.

Mention in current section 4.2 (study criteria) that intervention studies were excluded. This is mentioned later, but should also appear in the exclusion list. This is also where the reasons for exclusion “other (n=13)” from the PRISMA flowchart could be listed if not added to the figure.

- We added to the exclusion criteria that we excluded any interventional study. We have added the other reasons to the text of the figure.

  1. 23, current section 4.4, line 408 – list some of the additional confounders that some studies adjusted for (as examples).

- We thank reviewer 1 for the suggestion. We have added to examples to the additional confounders (education and marital status).

The conclusion section (p. 23) should include a bit more information (more like the abstract) in specifically what results were found (a brief overview of the general findings). In addition, there is a need for longitudinal studies including the periconceptual period to see if changes in maternal sleep from the periconceptual period to birth result in different maternal or child outcomes. This would clarify whether the periconceptual period was “vital” for later health & development, or whether adversity during this period could be overcome by later prenatal experiences.

- The reviewer makes good points on extending the conclusion section and mentioning the importance of longitudinal studies. We have made the following changes to address those points:

We have added the following sentence to provide more information: “... such as a higher risk of maternal hypertensive disorders and of preterm birth, with the association between sleep problems and maternal mood disorders being the most consistent.”

We have added the following sentence to the conclusion: “Given the high prevalence, future research should continue to study the short-term and long-term effects of sleep problems in the periconceptional period, preferably in a longitudinal design. In a design like this, starting from the preconceptional period and including pregnancy and the postpartum period, it would be possible to study whether sleep problems in the preconceptional period have a large effect on both mother and offspring or whether these effects can be overcome by later antepartum and postpartum experiences.”

Reviewer 2 Report

In this systemic review, the authors assessed maternal sleep problems during the periconceptional period—a period often overlooked in studies investigating sleep problems during pregnancy—and found that it is associated with a higher risk of various adverse outcomes in both mother and offspring. This is a well-designed study. Rigorous methods were implemented to shortlist relevant research articles, and appropriate statistical tests were performed. The outcome has clinical relevance as it identifies that paying attention to maternal sleep and treating sleep problems even before conception is essential for the health of both mother and the future offspring. The authors also honestly and explicitly explain the limitations of the study.

  • In study characteristics, the authors mention that the study includes data from 229067 participants. From table 1, it seems some studies (e.g., Facco et al., 2018 and 2019) used data from the same participants in different studies. Moreover, different studies examined different parameters in subjects. Therefore, I think listing the total number of participants the way it is now can be misleading.
  • How is the quality score determined?
  • The association of sleep problems with mood disorders was very consistent. What could be the possible physiological/neurophysiological/hormonal changes underlying the observations? Is there any such evidence from human/animal studies?

Author Response

In this systemic review, the authors assessed maternal sleep problems during the periconceptional period—a period often overlooked in studies investigating sleep problems during pregnancy—and found that it is associated with a higher risk of various adverse outcomes in both mother and offspring. This is a well-designed study. Rigorous methods were implemented to shortlist relevant research articles, and appropriate statistical tests were performed. The outcome has clinical relevance as it identifies that paying attention to maternal sleep and treating sleep problems even before conception is essential for the health of both mother and the future offspring. The authors also honestly and explicitly explain the limitations of the study.

- First, we would like to thank reviewer 2 for their kind words. We appreciate the amount of work that has been done reviewing our paper.

In study characteristics, the authors mention that the study includes data from 229067 participants. From table 1, it seems some studies (e.g., Facco et al., 2018 and 2019) used data from the same participants in different studies. Moreover, different studies examined different parameters in subjects. Therefore, I think listing the total number of participants the way it is now can be misleading.

- We recognize the comments made by reviewer 2. Two papers by Facco et al. studied indeed the same population. On top of that, Nakahara et al. (2020) and (2021) studied the same population, and Haney et al. (2014) and Okun et al. (2013) analyzed the same participants. Therefore, we need to subtract 7,524, 73,827, and 160 participants from the total of 229,067 participants, which adds up to a total of 148,096. We added the following sentence to line 72 in the methods section: “However, since a number of studies analyzed the same population, a total of 148,096 unique participants were studied for 12 different outcomes.”

How is the quality score determined?

- The quality score is determined through the ErasmusAGE criteria, which is explained in section 2.5. The score was designed based on previously published scoring systems (Carter et al. (2010) and the Quality Assessment Tool for Quantitative Studies). The quality score is composed of 5 items (study design, study size, exposure, outcome, adjustments), and each item is allocated 0, 1 or 2 points. This allows a total score between 0 and 10 points, 10 representing the highest quality. Since 2 points are given for interventional studies (criterion study design), and we excluded those, 9 points was the maximum score possible.

The association of sleep problems with mood disorders was very consistent. What could be the possible physiological/neurophysiological/hormonal changes underlying the observations? Is there any such evidence from human/animal studies?

- The association seems to be the most consistent indeed. There is, both during pregnancy but also outside of this window, a strong reciprocal relation between sleep and mood disorders. Sleep problems can precede depressive symptoms and are also part of the diagnosis, but depressive symptoms can also cause sleep problems. Therefore, a causal relation is difficult to prove. One of the possible underlying hormonal changes may be cortisol. Changes in cortisol levels, whether these are too high or too low, are associated with various mood disorders, but may also affect sleep. This has been briefly described in section 3.3 (lines 358-360). However, based on this review, we cannot draw any strong conclusions on this.